# Increasing the Accuracy of Free-Form Surface Multiaxis Milling

**DOI:** 10.3390/ma14010025

**Published:** 2020-12-23

**Authors:** Marek Sadílek, Zdeněk Poruba, Lenka Čepová, Michal Šajgalík

**Affiliations:** 1Department of Machining and Assembly, Faculty of Mechanical Engineering, VŠB-Technical University of Ostrava, 708 33 Ostrava, Czech Republic; lenka.cepova@vsb.cz; 2Department of Applied Mechanics, Faculty of Mechanical Engineering, VŠB-Technical University of Ostrava, 708 33 Ostrava, Czech Republic; zdenek.poruba@vsb.cz; 3Department of Machining and Production Technologies, University of Zilina, 010 26 Zilina, Slovakia; michal.sajgalik@fstroj.uniza.sk

**Keywords:** multi axis milling, 5-axis milling, accuracy, CAM system

## Abstract

This contribution deals with the accuracy of machining during free-form surface milling using various technologies. The contribution analyzes the accuracy and surface roughness of machined experimental samples using 3-axis, 3 + 2-axis, and 5-axis milling. Experimentation is focusing on the tool axis inclination angle—it is the position of the tool axis relative to the workpiece. When comparing machining accuracy during 3-axis, 3 + 2-axis, and 5-axis milling the highest accuracy (deviation ranging from 0 to 17 μm) was achieved with 5-axis simultaneous milling (inclination angles β_f_ = 10 to 15°, β_n_ = 10 to 15°). This contribution is also enriched by comparing a CAD (Computer Aided Design) model with the prediction of milled surface errors in the CAM (Computer Aided Manufacturing) system. This allows us to determine the size of the deviations of the calculated surfaces before the machining process. This prediction is analyzed with real measured deviations on a shaped surface—using optical three-dimensional microscope Alicona Infinite Focus G5.

## 1. Introduction

In Figure 1 there is a schematic diagram of the aspects that should be monitored when comparing the machining process during 3-axis and multiaxis milling.

The tool axis inclination against a workpiece has a significant influence on the size and direction of the cutting forces [1,2] (it means the individual cutting force components). The radial, axial, and tangential cutting forces tend to push the tool apart. It is, therefore, appropriate to verify to what extent the proposed changes in the tool axis inclination angle affect the accuracy of the machined surface.

Studies on optimizing the cutting conditions and their influence on cutting forces are described in [3,4,5,6,7]. Studies on machined surface topography while changing the orientations of the tool axis can be found in the literature [8,9,10,11,12,13,14]. Studies on the contact of the tool with the workpiece are described in [3,7,13]. The deformation deflection of the cutting tools or the deformation of the machined parts are mentioned in [1,8,14,15]. The problem of the durability of the cutting tool is described in the literature [16,17,18]. Shape and chip geometry is described in the literature [3,4].

In the literature [19,20], errors caused by cutter deflection when machining a sculptured part using a ball-end milling tool are described. In the literature [20], a flexible model for estimating the form error in three-axis ball-end milling of the sculptured surface can be found.

Using multiaxis machining (3 + 2- and 5-axis machining) includes these benefits, which is based on previous research by the authors [2,14,21,22,23,24]:increasing milling accuracy is proved in this article,decreasing surface roughness in the pick feed direction and the feed direction,decreasing the cutting time (using a bigger ae, fz with the same surface roughness),constant cross-sectional area of the chip,increasing durability (tool life) of the cutting tool,constant cutting conditions can be used and cutting speed can be increased,decreasing the size of the cutting forces components,a favorable orientation of cutting force direction,increasing the functional surface properties of the machined surface,decreasing of the cutting temperature and inhibition of the self-excited oscillations.

## 2. Kinematics of the Milling Strategy 

The problem of 3-axis and multi-axis milling with reference to the kinematics and position of the tool axis of the milling cutter is described in the author’s articles [2,14,22], see Figure 2.

The possibilities of tool inclination towards a normal surface are shown in Figure 2.

There are two ways that can be used for an inclination in the feed direction (see Figure 3) as well as for an inclination that is perpendicular to the feed direction, see Figure 4 [2,14].

The literature [14,22,25] shows how changing the tilting of the tool affects the effective cutter diameter and then the effective cutting speed.

The scientific literature describes using the tool axis inclination angle at the range from 10 to 30°. Another scientific paper uses both inclination angles at 15°. References [8,9] recommend a range from 10 to 20°.

## 3. Experimental Work

The experimental part describes the machined accuracy after machining with different positions of the tool axis. It means a tool axis relative to the normal surface. The original CAD model was created in CAM system Inventor.

The dimensions of the model (see Figure 5) were designed with regard to the use of the finishing tool (a 6 mm diameter ball-end milling cutter) and its subsequent effective measurement on the measuring instrument.

The experiment consists of three parts, see Figure 6:(a)3-axis milling—the tool axis inclination angles are β_f_ = 0°, β_n_ = 0°, i.e., without changing the tool axis angle relative to the workpiece. It consists of one sample.(b)3 + 2-axis milling—constant tilt relative to the orientation of the surfaces, β_f_—pulled tool, β_n_—tilt in pick feed direction. It consists of 24 samples.(c)5-axis—simultaneous movement (X, Y, Z, B, C), tool axis inclination angles follow the orientation of the surface—the slope of the tool axis, versus the normal to the surface, varies depending on the shape of the model surface. It consists of 25 samples.

Each sample of these three parts (each setup) was machined and measured three times, but we are presenting only representative samples.

In the experiment, 3 + 2-axis machining is 3-axis machining with fixed toll axis orientation. Simultaneous 5-axis machining moves the cutting tool on the x, y, and z axes and rotates the A, B, and C axes (in our kinematic B and C axes) to maintain continuous contact between the tool and workpiece, unlike 3 + 2-axis machining, where the part is in a fixed orientation.

Five positions (tilts) of the tool in the feed direction and five positions in the pick feed direction were analyzed. Changing the orientation of the axis in the feed direction by β_f_ = 0° to β_f_ = 20° was partitioned at 5°. Changing the orientation of the axis in the pick feed direction by β_n_ = 0° to β_n_ = 20° was partitioned at 5°. Maximum inclination angle was set according to the shallow surfaces of the sample. Each sample has a size of 40 mm × 40 mm × 25 mm. In the experiments, a 5-axis milling machine was used. 

The technological parameters of the cutting process were adopted as a constant for each of the machined samples (see Table 1 and Table 2).

Specified cutting conditions and their relationships are shown in Figure 7. 

It is important to note that a cycle was used (in the Heidenhain control system) with a tolerance 0.01 mm, which affects the machining accuracy, surface roughness, and feed rate. Cycle 32 was used, including a tolerance TA for rotation axes in the control system Heidenhain iTNC 530:CYCL DEF 32.0 TOLERANCE (using special cycles in Heidenhain iTNC),CYCL DEF 32.1 T0.01 (the tolerance was set at 0.01 mm),CYCL DEF 32.2 HSC-MODE:1 TA1 (MODE: 1 means for finishing, a permissible deviation of the position of the rotary axes TA = 1°).

It is suitable to use CYCL DEF 32.0 TOLERANCE during 5-axis simultaneous machining, including a tolerance for rotary axes TA. TA is the permissible deviation of the position of the rotary axes with an active M128. The TNC reduces the feed in such a way that the slowest axis travels as a maximum feed (during multiaxis machining). Rotational axes are usually slower than linear axes. By setting a larger tolerance (for example, TA = 10°) the machining time for multi-axis machining can be considerably decreased because the TNC does not always need to travel the rotary axis to the preset target position. The contour does not interfere with setting this tolerance (1°) of the rotary axes. Only the position of the rotary axis to the workpiece surface is changed (ranging 1°).

### 3.1. Accuracy Measurement

Machined sample measurements were carried out with an optical three-dimensional microscope Alicona Infinite Focus G5, (Alicona Imaging GmbH, Raaba/Graz, Austria), see Figure 8. The optical 3D micro coordinate measurement system is suitable for accuracy and surface roughness measurements.

The samples measured using this device were compared to the default model created in the CAD system Inventor. These real surfaces were also compared with the predicted surfaces calculated in the CAM system Mastercam. The models were saved with the input format necessary for Alicona Infinite Focus G5 (software IF MeasureSuite, Alicona Imaging GmbH, Raaba/Graz, Austria) in the format *. stl with a tolerance of 0.0001 mm (so as not to affect the measurement results).

Real machined surface is presented in Figure 9.

The accuracy comparison was mainly focused on the problematic radius crossing of surfaces at the area of the center of the workpiece. Inaccuracies were found mostly at this area in 3-axis and 3 + 2-axis milling.

A range of the variance scale was selected at −50 to +50 μm for all samples. Machined surfaces colored light green have minimal deviations (range of 10 to 20 μm), see Figure 10. A range from 20 to 40 μm deviations is in the central area of the workpiece. 

The sample machined with a 3 + 2-axis milling with a tool inclination β_f_ = 20°, β_n_ = 0° shows the best machining results from the 3-axis and 3 + 2-axis milling group of samples, see Figure 11. Deviations 0 to 20 μm were achieved at the center of the workpiece.

The sample machined with a 3 + 2-axis milling with a tool inclination β_f_ = 10°, β_n_ = 20° shows the worst machining results, see Figure 12. Deviations in the center of the workpiece reach values higher than the upper limit of the selected range of 50 μm.

By using simultaneous 5-axis milling, the machining accuracy of all samples was improved. The problematic area in the center of the samples is more accurately machined by simultaneous 5-axis milling compared to 3-axis and 3 + 2-axis milling, where deviations are significantly unsymmetrically changing.

The sample machined using a 5-axis milling with a tool inclination β_f_ = 0°, β_n_ = 10° shows the best machining results from all groups (3-axis, 3 + 2-axis, and 5-axis milling), see Figure 13. A deviation ranging from 0 to 17 μm was measured at the center of the workpiece.

The deviations were measured at a different area during the 5-axis milling compared to the 3-axis and 3 + 2-axis milling, see Figure 14 (area A marked on the model). The size of the deviations in this area (a slight convex curvature on the right side of the sample—area A marked on the model) differs for each 5-axis machining, see Figure 13. This phenomenon has not been observed during 3 + 2-axis milling.

### 3.2. The Prediction of Milled Surface Errors in the CAM System

The CAM system allows comparing the CAD model with the prediction of milled surface errors. This allows determining the size of errors of the calculated surfaces before machining. A tolerance was set at 0.001 mm (path tolerance, part tolerance, tool shape tolerance). Figure 15 shows the verification procedure of the machining accuracy.

Figure 16 shows a comparison of a CAD model (made in the CAD system) and the prediction of milled surface errors using 3-axis milling in the CAM system—Mastercam. When comparing the prediction of milled surface errors after a 3-axis milling in the CAM system, see Figure 16, and the measurement result on the Alicona, see Figure 10, a conformity can be seen in the problematic area in the center of the sample (the predicted deviations of accuracy are 20 to 40 μm).

Figure 17 shows two comparisons of the CAD model from Inventor and a prediction of milled surface errors using a 3 + 2-axis milling in the CAM system.

During changing the tool axis inclination angle (3 + 2-axis milling), the calculated shape of machined surfaces in the CAM system does not change significantly, see Figure 18. The individual tilts have very little effect on the shape of the calculated workpiece in the CAM system. 

There is a similar result when comparing the calculated residual material in the CAM system and the real results of the measurement in the problematic area of the center of the sample, see Figure 18. The prediction of the formation of a planar surface (on the left of the sample) in the CAM system Mastercam does not correspond to the actually measured results. This prediction of 3 + 2 milling before the real process is not conclusive in practice. This comparison is, therefore, rather indicative.

Figure 19 shows a comparison of the CAD model and the prediction of milled surface errors using a 5-axis milling in the CAM system. These errors are 0 to 15 µm. 

There are very similar results on all surfaces when comparing the predicted milled surface errors and the real errors using 5-axis milling. 5-axis milling has the highest consistency of the predicted errors with the measured errors.

### 3.3. Surface Roughness Measurement

Surface roughness was measured only in problematic areas in the center of the sample, see Figure 9.

Measurement has been performed according the standard ISO 25178-1 [26] and ISO 25178-2 [27]. Filtration L-filter Lc (cut off) has been set according the standards with filter λc = 800 µm. 

Table 3 and Table 4 show maximum deviation and selected surface roughness parameters in the center area of samples (Sa—Arithmetical mean height of the scale-limited surface, Sp—Maximum peak height of the scale-limited surface, Sz—Maximum height of the scale-limited surface, and Vmp—Peak material volume of the scale-limited surface). Parameter Vmp is one of many volume parameters of surface roughness and gives better knowledge about the quality of the rated surface. It can be clearly seen in Table 4. Where green are positive results and red negative results. A darker green color means better positive results. A darker red color indicates worse negative results. The results of the previous author’s studies show that the surface roughness is influenced by the inclination of the tool [2,14,21]. Surface roughness can affect accuracy [28,29,30,31,32,33,34,35,36]. Selected roughness parameters compare the peaks that can affect geometric accuracy. 

Better surface roughness was achieved with 3 + 2-axis milling. Selected surface roughness parameters show a lower range of values (max. value—min. value) during 3 + 2-axis machining. 3 + 2-axis has better results than 5-axis. However, the values of the surface roughness are very similar. Within the extended uncertainty of measurement, values are close to each other, in particular the parameter Sa (±0.15 µm and ±0.51 µm).

Figure 20 shows the advantages of the tilting tools for achieving an improved machined surface. Stable and best results have inclination β_f_ = 15° in the range from β_n_ = 10 to 20° during 3 + 2-axis milling. 

## 4. Discussion and Conclusions

It can be concluded that the tool inclination has an influence on the accuracy of machining after the final experiment evaluation. This paper shows that using different milling strategies during the ball-end milling finishing of free-form surfaces has an effect on accuracy and surface errors. When comparing machining accuracy during 3-axis, 3 + 2-axis, and 5-axis milling (under the same conditions—the same cutting conditions (the same f_z_, v_c_, a_p_, a_e_, etc., see Table 2), the highest accuracy was achieved with 5-axis simultaneous milling (β_f_ = 10 to 15°, β_n_ = 10 to 15°, deviation range from 0 to 17 μm). Inaccuracies were found mostly at problematic areas of the center in 3-axis and 3 + 2-axis milling (deviations range from 0 to 56 μm), see Table 4. The arithmetic mean of inaccuracies in the 5-axis machining is 18 µm and in the 3 + 2-axis machining is 47 µm.

The prediction (calculation) of residual material in the CAM system for a group of samples machined by 5-axis milling is the most consistent with the real-measured residual material. The results show the quality of the computational algorithm used in the CAM module of 5-axis milling.

It is beneficial to use tool axis inclination angle in the tool feed direction and avoid inclination β_f_ = 0°, where are surface roughness parameters Vmp and Sz, Sp highest. Where in the axis the material is pushed not cut, the effective cutting speed is 0 m·min^−1^. We recommend using tool axis inclination angle in the interval β_f_ = 15 to 20°, β_n_ = 5 to 20°. 3 + 2-axis milling shows more stable values (lower range of values) of surface roughness than 5-axis simultaneous milling. Free-form finishing milling with a targeted tool axis inclination angle can improve surface roughness or can replace the grinding operation, i.e., the manual finishing grinding operation of free-form surfaces.

## Figures and Tables

**Figure 1 materials-14-00025-f001:**
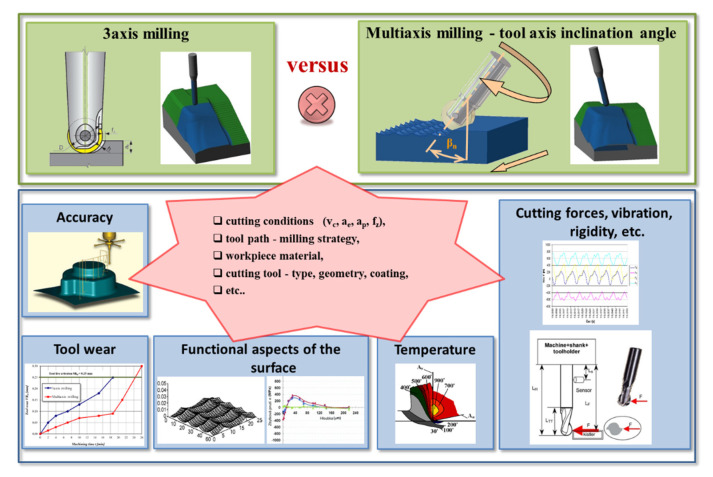
Aspects of the machining process.

**Figure 2 materials-14-00025-f002:**
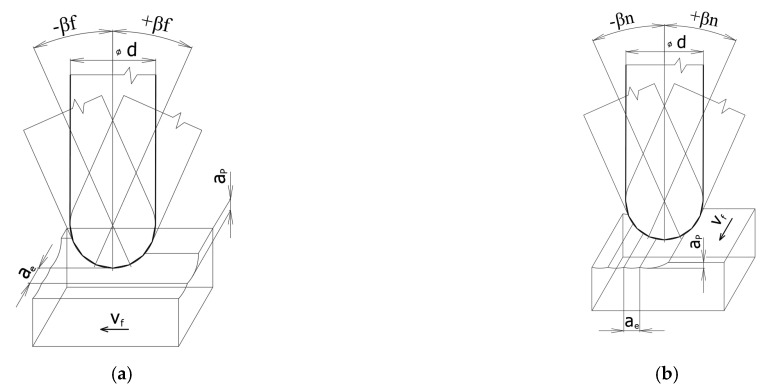
The possibilities of the milling strategy with a tool axis inclination angle [2,14]. (**a**) Tilt in feed direction, (**b**) tilt in pick feed direction (d—tool diameter, a_p_—axial depth of cut, a_e_—radial depth of cut, v_f_—feed direction, β_f_—tool axis inclination angle in the feed direction, β_n_—tool axis inclination angle in the pick feed direction.

**Figure 3 materials-14-00025-f003:**
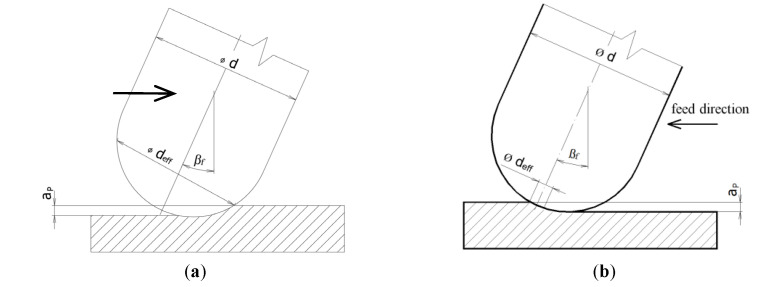
Tool axis inclination angle β_f_ in feed direction [2,14] (**a**) pulled tool (**b**) pushed tool (d—tool diameter, d_eff_—effective tool diameter, a_p_—axial depth of cut).

**Figure 4 materials-14-00025-f004:**
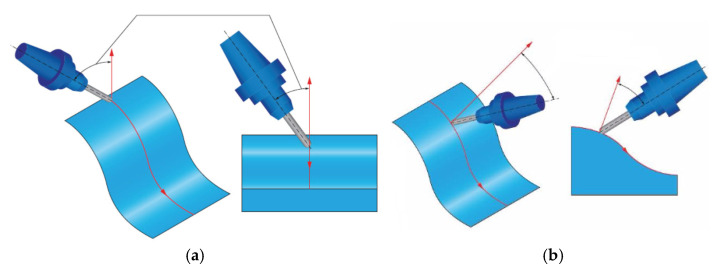
Tool axis inclination angle in both directions (**a**) Tool axis inclination angle in pick feed direction β_n_ (**b**) Tool axis inclination angle in feed direction β_f_.

**Figure 5 materials-14-00025-f005:**
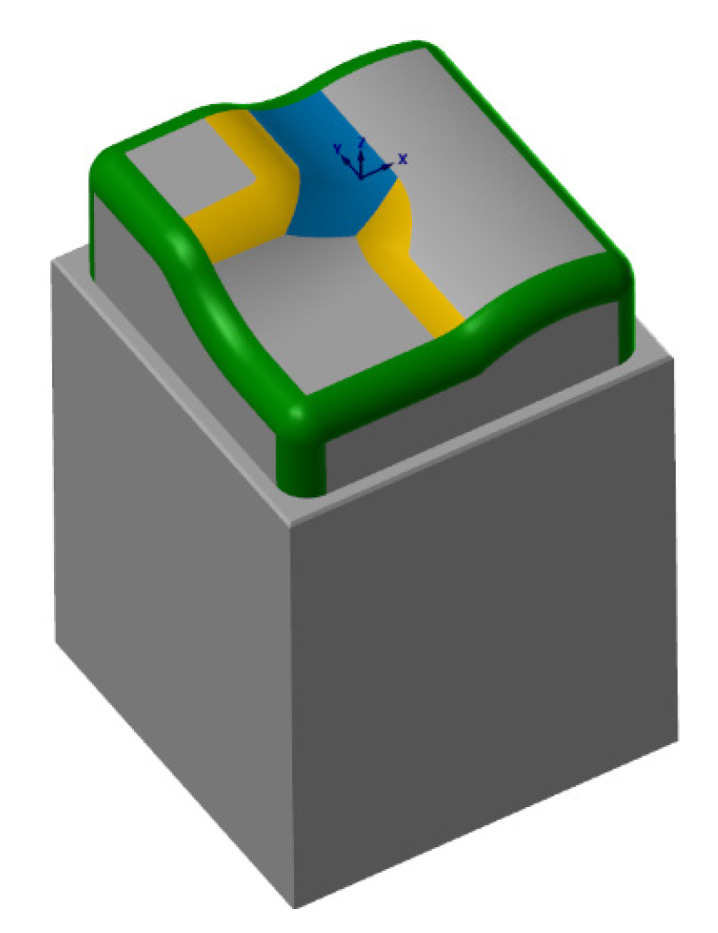
CAD geometry used in experiments (Surfaces: R2 mm—green, R5 mm—yellow, R6 mm—blue).

**Figure 6 materials-14-00025-f006:**
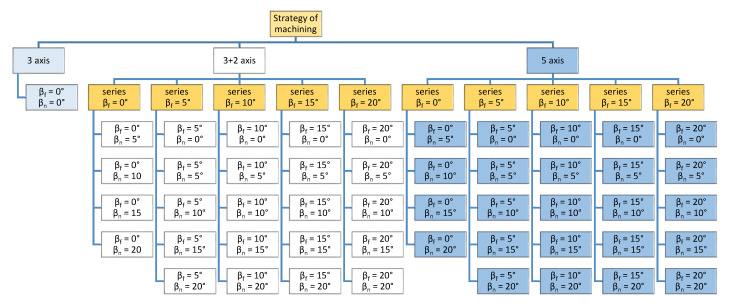
Chart of milling samples.

**Figure 7 materials-14-00025-f007:**
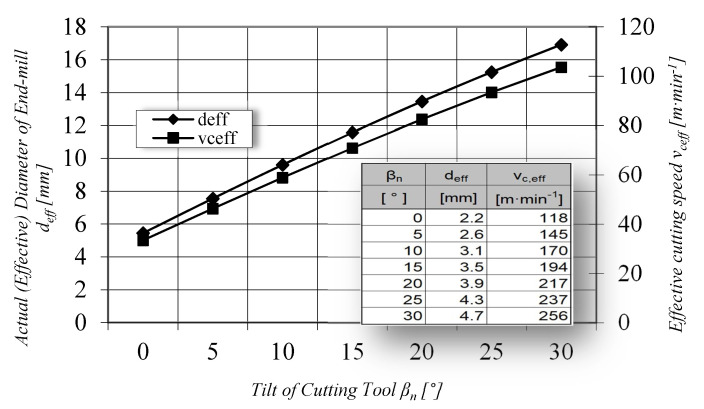
The dependence of the effective cutting speed v_ceff_ and the effective tool diameter d_eff_ at the tool axis inclination angle β_n_, (d = 6 mm, a_p_ = 0.2 mm, v_c_ = 330 m·min^−1^, n = 17,500 min^−1^).

**Figure 8 materials-14-00025-f008:**
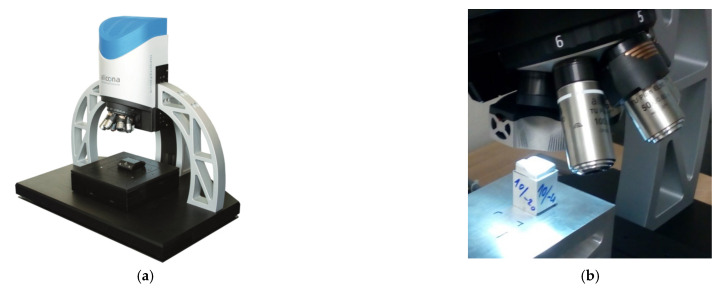
3D micro coordinate measurement system (**a**) Alicona Infinite Focus G5 (**b**) sample for measurement.

**Figure 9 materials-14-00025-f009:**
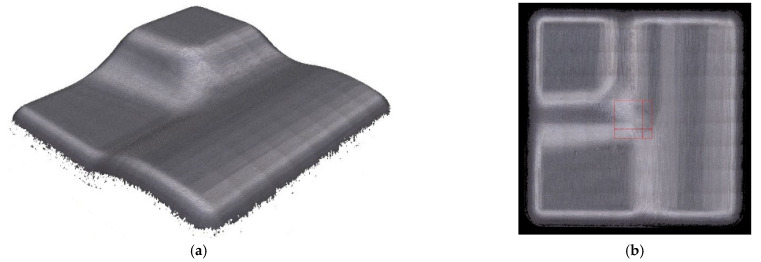
An example of the result of the scanned surface of a single sample on a Alicona Infinite Focus G5 (**a**) 3 + 2 milling, tool axis inclinations β_f_ = 5°, β_n_ = 10°, (**b**) red square area (4 mm × 4 mm) for surface roughness measurement.

**Figure 10 materials-14-00025-f010:**
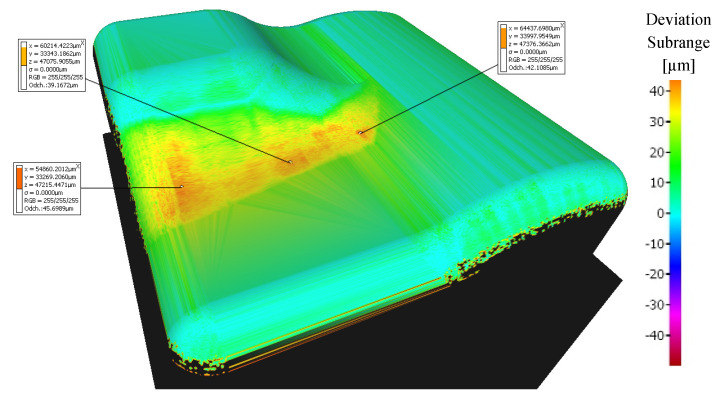
Comparison of the initial model with the workpiece after 3-axis milling (without tool axis inclination: β_f_ = 0°, β_n_ = 0°).

**Figure 11 materials-14-00025-f011:**
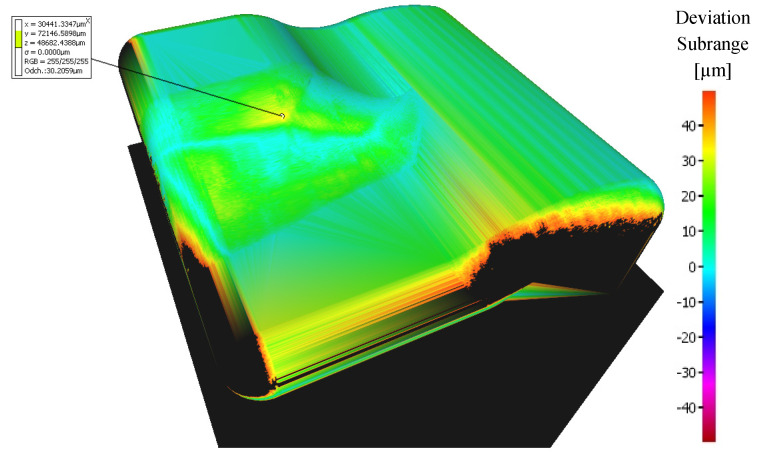
Comparison of the initial model with the workpiece after 3 + 2-axis milling (tool axis inclination: β_f_ = 20°, β_n_ = 0°).

**Figure 12 materials-14-00025-f012:**
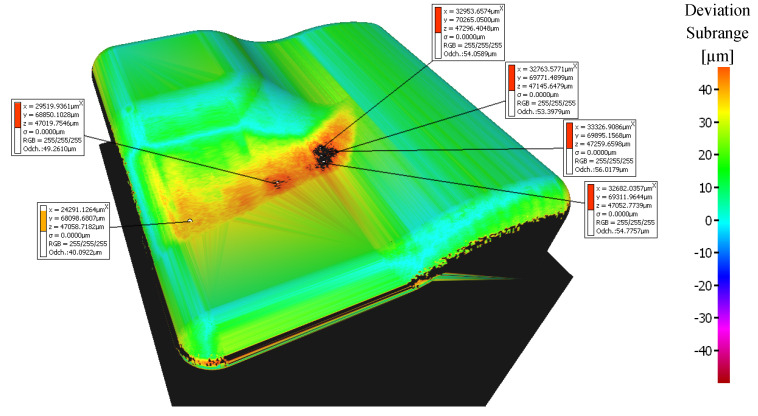
Comparison of the initial model with the workpiece after 3 + 2-axis milling (tool inclination β_f_ = 10°, β_n_ = 20°).

**Figure 13 materials-14-00025-f013:**
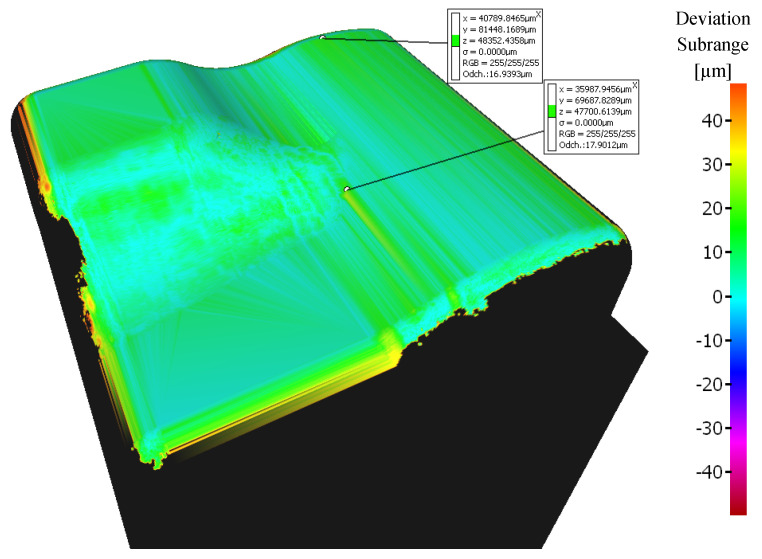
Comparison of the initial model with the workpiece after 5-axis milling (tool inclination β_f_ = 0°, β_n_ = 10°).

**Figure 14 materials-14-00025-f014:**
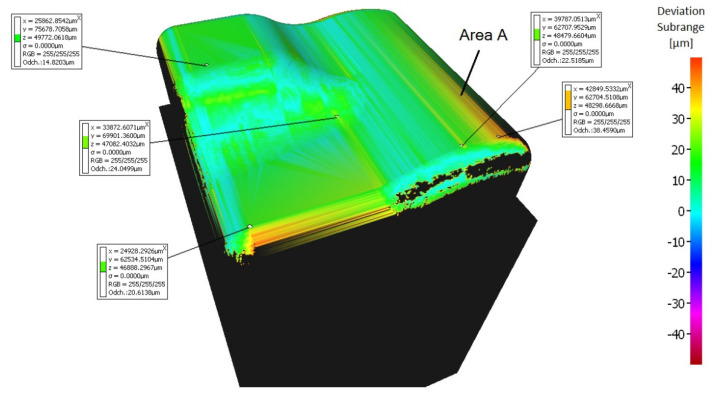
Comparison of the initial model with the workpiece after 5-axis milling (tool inclination β_f_ = 15°, β_n_ = 5°).

**Figure 15 materials-14-00025-f015:**
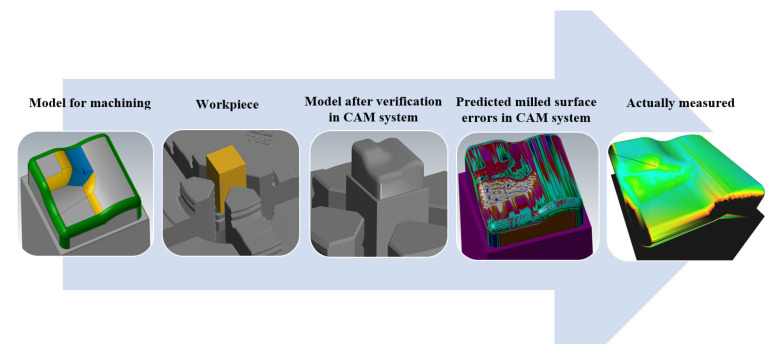
Diagram of the verification procedure of the machining accuracy.

**Figure 16 materials-14-00025-f016:**
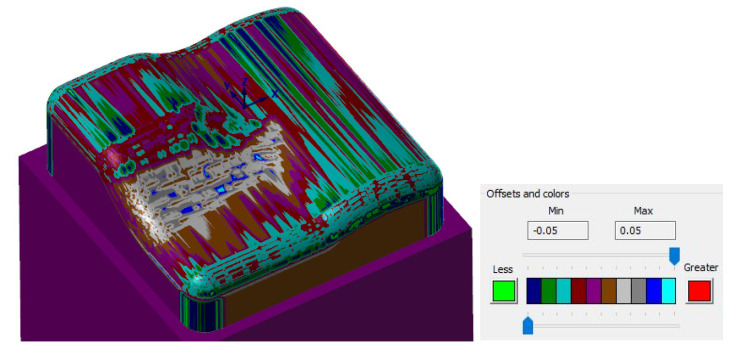
Comparison of CAD model (Inventor) and prediction of milled surface errors using 3-axis milling (Mastercam).

**Figure 17 materials-14-00025-f017:**
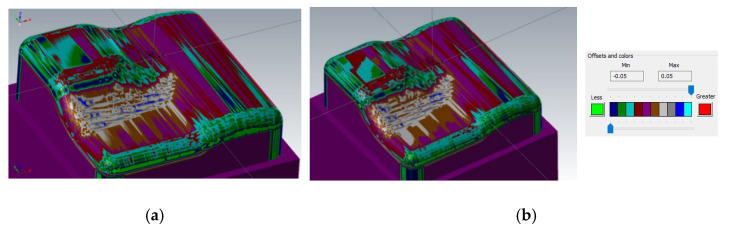
Comparison of the CAD model (Inventor) and the prediction of milled surface errors using 3 + 2-axis milling (Mastercam), (**a**) tool inclination β_f_ = 20°, β_n_ = 0°, (**b**) tool inclination β_f_ = 10°, β_n_ = 20°).

**Figure 18 materials-14-00025-f018:**
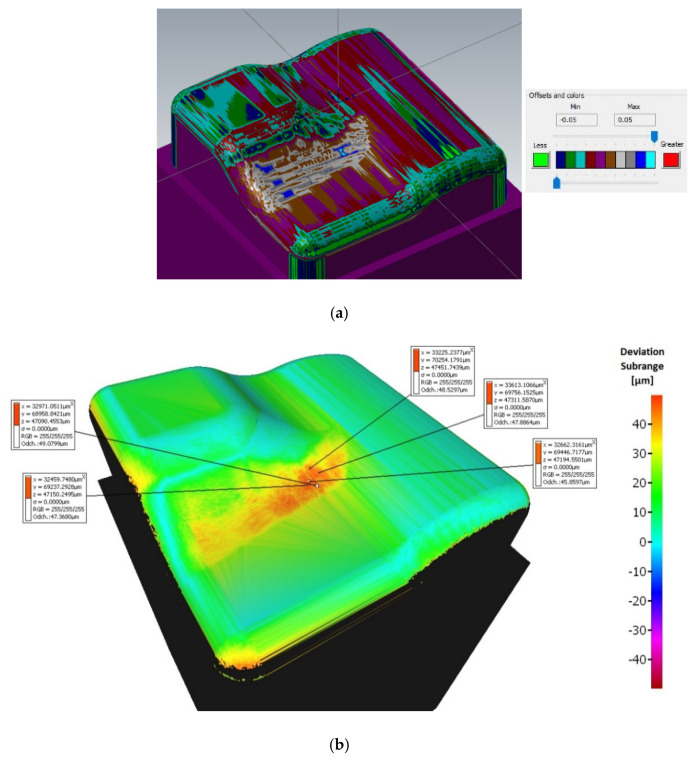
Comparison of predicted surfaces errors and real surface errors, 3 + 2-axis milling, tool inclination β_f_ = 15°, β_n_ = 5°, (**a**) predicted errors and (**b**) real surface errors.

**Figure 19 materials-14-00025-f019:**
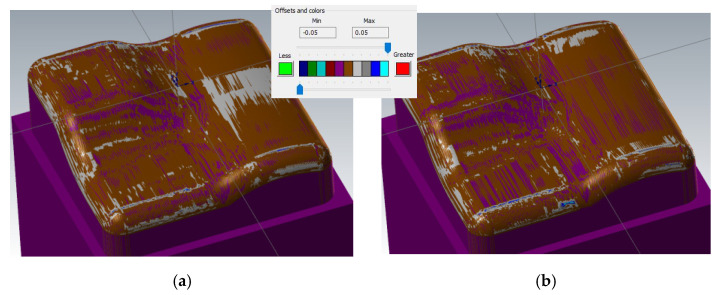
Comparison of the CAD model (Inventor) and the prediction of milled surface errors using 5-axis milling (Mastercam), (**a**) tool inclination β_f_ = 5°, β_n_ = 10° and (**b**) tool inclination β_f_ = 10°, β_n_ = 10°).

**Figure 20 materials-14-00025-f020:**
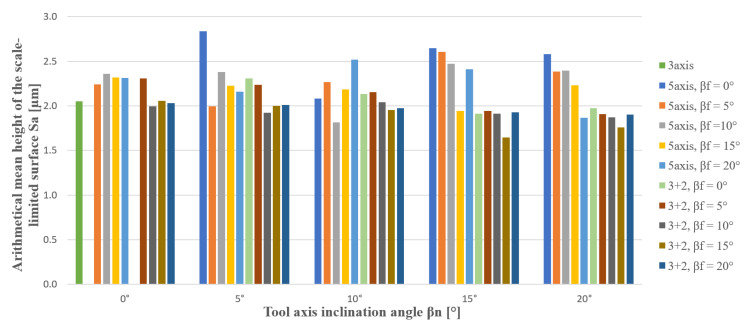
Surface roughness (Sa—Arithmetical mean height of the scale-limited surface) dependence on tool axis inclination angle.

**Table 1 materials-14-00025-t001:** Technological parameters of the cutting process.

Machine	DMG MORI DMU 50, DMG MORI. CO., LTD., Wernau, Germany	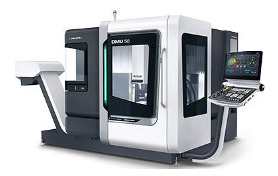
Control system	Heidenhain iTNC 530, HEIDENHAIN CORPORATION, Schaumburg, IL, USA	
CAM system	Mastercam 2017	
Material of workpiece	Aluminum alloy EN AW-6060—AlMgSi0,5 F19	
Workpiece	40 mm × 40 mm × 25 mm	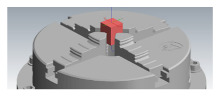
Clamping	Four-jaw chuck on the column	
Tool	Solid carbide end mill, CoroMill^®^ Plura, D6 mm, Sandvik AB, Stockholm, Sweden	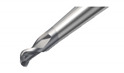
Tool holder	High-precision hydraulic chuck—CoroChuck 930, Sandvik AB, Stockholm, Sweden	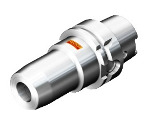
Tool overhang	40 mm	

**Table 2 materials-14-00025-t002:** Cutting conditions of finishing operation.

Ball Endmill Diameter	Spindle rev.	Cutting Speed	Axial Cutting Depth	Radial Cutting Depth	Feed Per Tooth	Feed	Plunge and Retract Feed
d	n	v_c_	a_p_	a_e_	f_z_	f	f_p_, f_r_
[mm]	[min^−1^]	[m·min^−1^]	[mm]	[mm]	[mm]	[mm·min^−1^]	[mm·min^−1^]
6	17,500	330	0.2	0.08	0.08	2800	1910
Colling			Blasocut 2000 CF, Art. 875-12				
Strategy			tool path “zig zag”, a conventional milling and climb milling combination, the tool axis inclination angle in both directions—feed direction β_f_ and pick feed direction β_n_—called in practice the “pulled tool”.				

**Table 3 materials-14-00025-t003:** Maximum deviation and surface roughness during 3-axis machining.

Inclination Angle	Max. Deviation	Surface Roughness			
**β_f_, β_n_**		**Sa**	**Sp**	**Sz**	**Vmp**
[°]	[µm]	[µm]	[µm]	[µm]	[mL·m^−^²]
0°0°	42	2.05	13.06	25.72	0.148
Extended uncertainty	3.38	0.11	1.00	2.41	0.081

**Table 4 materials-14-00025-t004:** Maximum deviation and surface roughness during 3 + 2-axis and 5-axis machining (where green are positive results and red negative results).

	Max. Deviation		Surface Roughness							
Inclination Angle β_f_, β_n_	3 + 2 -axis	5-axis	3 + 2 -axis Sa	5-axis Sa	3 + 2 -axis Sp	5-axis Sp	3 + 2 -axis Sz	5-axis Sz	3 + 2 -axis Vmp	5-axis Vmp
[°]	[µm]	[µm]	[µm]	[µm]	[µm]	[µm]	[µm]	[µm]	[mL·m^−^²]	[mL·m^−^²]
0°, 5°	38	14	2.31	2.84	22.64	18.56	38.24	46.95	0.159	0.193
0°, 10°	42	18	2.13	2.08	19.81	21.39	35.74	37.93	0.160	0.139
0°, 15°	47	12	1.91	2.65	11.06	34.50	25.40	59.81	0.122	0.190
0°, 20°	45	12	1.98	2.58	18.45	22.79	37.16	44.43	0.125	0.175
5°, 0°	38	19	2.31	2.24	22.64	15.65	40.96	28.22	0.192	0.139
5°, 5°	50	5	2.24	1.99	17.96	18.82	41.57	38.13	0.154	0.158
5°, 10°	49	10	2.15	2.27	15.07	15.64	33.83	29.71	0.145	0.165
5°, 15°	52	15	1.94	2.61	12.58	22.52	35.57	46.85	0.122	0.180
5°, 20°	51	10	1.91	2.38	17.34	14.08	40.41	29.26	0.126	0.147
10°, 0°	43	15	2.00	2.36	18.37	15.37	40.41	31.87	0.160	0.155
10°, 5°	52	20	1.93	2.38	13.87	26.77	31.64	44.77	0.140	0.167
10°, 10°	53	10	2.04	1.82	15.04	16.71	39.57	30.27	0.150	0.130
10°, 15°	55	21	1.91	2.47	16.62	19.73	35.67	39.22	0.127	0.160
10°, 20°	56	25	1.87	2.40	17.36	15.55	32.81	35.32	0.113	0.174
15°, 0°	42	19	2.06	2.32	14.61	26.24	29.05	38.80	0.150	0.149
15°, 5°	48	24	2.00	2.23	12.88	16.18	28.00	36.16	0.125	0.158
15°, 10°	46	29	1.95	2.18	15.24	15.02	33.26	28.75	0.129	0.151
15°, 15°	56	9	1.64	1.94	9.88	14.41	21.65	27.14	0.093	0.118
15°, 20°	43	24	1.76	2.23	12.31	13.85	23.54	31.17	0.100	0.145
20°, 0°	20	31	2.03	2.31	15.41	15.91	30.67	31.57	0.127	0.151
20°, 5°	46	28	2.01	2.16	21.68	19.94	42.57	34.83	0.157	0.140
20°, 10°	53	27	1.97	2.52	16.07	34.46	33.81	54.73	0.151	0.193
20°, 15°	51	32	1.93	2.41	13.83	17.39	28.24	32.13	0.139	0.158
20°, 20°	52	10	1.90	1.86	12.05	11.64	30.85	23.94	0.109	0.112
Range of values	0 to 56	0 to 32	0.67	1.02	12.77	22.86	20.92	35.87	0.099	0.081
Arithmetic mean	47	18	1.99	2.30	15.95	19.30	33.78	36.75	0.136	0.156
Extended uncertainty	15.68	15.69	0.32	0.51	7.01	12.10	11.75	17.97	0.083	0.082

## Data Availability

The data presented in this study are available on request from the corresponding author.

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
