# Peer review of "Increasing the Accuracy of Free-Form Surface Multiaxis Milling"

_materials, 2020, doi:10.3390/ma14010025_

Round 1
Reviewer 1 Report
This paper proposes the analysis of the machining precision of parts using 3-axis, 3+2 axis, and 5-axis machining method by considering inclination angles. The testing results reveal that 5-axis machining has the best machining quality with the setup of βf = 0° and βn = 10°. This research provides a useful guide on the selection of tool inclination angles for free form surface milling although the selection of inclination angle for machining may change with machining parameters, materials, experimental part or even the machine tools.
The comments are as follows:
[1] Section 1, line 46 to line 64, please correct the missing reference number.
[2] For the experimental work, one sample is used for 3-axis machining, 24 samples are used for 3+2 axis machining, and 25 samples for the 5-axis machine tool. Actually, for each setup, only one sample was tested. To get a reliable statistical result, more samples for each inclination angle are needed. For example, five or ten samples. In this case, you can get a strong conclusion.
[3] For the inclination angle setup, I did not see an explanation that why the maximum 20 degrees is selected. Herein, a strong reason is needed. For example, Chenwei Shan 2016 mentioned that when the tool inclination angles are 15° or 45°, both the milling force and the deformation of the test workpieces (Thin-walled freeform surface) are the smallest. So, a large inclination angle could be considered in this research.
[4] Did you calibrate the machine tool before the experiment? Calibration can make sure the experimental machine tool has a known accuracy state. Then, the original machine tool precision can not bring some affections to the final experiment result.
[5] Machine tools have different volumetric errors in the machining space. You need to know the volumetric error information for your machining space. Volumatic errors definitely can affect the comparison results.
[6] It is hard to recognize the detail of Fig. 11- Fig. 20, especially for some statistical parameters. You need to use figures with high solutions.
[7] Except for the roughness, parameters such as flatness and parallelism could also be selected for this research. You may consider these parameters.
[8] Could you please give the reader some explanations on the differences between 3+2 axis matching and 5-axis machining?
[9] Line 219, you mentioned that they are under the same condition, Why? For different setups, the G-code is different, or even the tool path is also different. Does this affect the result?
[10] Line 221-222, What is the result of 3-axis machining? Do you mean that 3-axis machining has a close machining quality as the 3+2 axis machining? This description is not clear. You can use a table to show the roughness of each sample.
Author Response
My Response to Reviewer
Dear Reviewer,
Thank you for your letter and the reviewers’ comments concerning our manuscript entitled. These comments are all valuable and very helpful for revising and improving our paper, as well as the important guiding significance to our researches.
We have studied comments carefully and have made correction which we hope meet with the approval. According to the comments, major changes in the revised manuscript are highlighted using the "Track Changes" function in Microsoft Word.
Comments and Suggestions for Authors reviewer 1
This paper proposes the analysis of the machining precision of parts using 3-axis, 3+2 axis, and 5-axis machining method by considering inclination angles. The testing results reveal that 5-axis machining has the best machining quality with the setup of βf = 0° and βn = 10°. This research provides a useful guide on the selection of tool inclination angles for free form surface milling although the selection of inclination angle for machining may change with machining parameters, materials, experimental part or even the machine tools.
The comments are as follows:
[1] Section 1, line 46 to line 64, please correct the missing reference number.
Missing reference number has been added. This sentence was corrected and improved: ”Using multiaxis machining (3+2 and 5axis machining) include these benefits, which is based on previous research by the authors [19], [20], [21], [35], [36]”.
[2] For the experimental work, one sample is used for 3-axis machining, 24 samples are used for 3+2 axis machining, and 25 samples for the 5-axis machine tool. Actually, for each setup, only one sample was tested. To get a reliable statistical result, more samples for each inclination angle are needed. For example, five or ten samples. In this case, you can get a strong conclusion.
We add: “Each sample of these three parts (each setup) was machined and measured fhree times, but we are presenting only representative samples (Fig.10 to Fig.15 and Fig.19).”
Yes, it is true, If we make setup ten times we can received more reliable statistical results. These experiments were time consuming and very costly, especially measuring took very long time and we tried to decrease finance aspects.
[3] For the inclination angle setup, I did not see an explanation that why the maximum 20 degrees is selected. Herein, a strong reason is needed. For example, Chenwei Shan 2016 mentioned that when the tool inclination angles are 15° or 45°, both the milling force and the deformation of the test workpieces (Thin-walled freeform surface) are the smallest. So, a large inclination angle could be considered in this research.
The reason for max. inclination angle 20° was respect to the shallow areas of the test sample. Higher inclinations for these surfaces are not suitable, due to the kinematics of the machine tool (movements). When there are large differences in the movement of the rotary axes at higher inclinations of the cutter when milling slightly inclined surfaces. Movements of rotary axes are much slower than movements of linear axes, resulting in an undesirable increase in machine time. A greater tilt of the tool axis (approaching 90 °) is applied to steep surfaces. Here we can use a strategy commonly named as constant z. In our research is used strategy - tool path “zig zag”.
Maximum inclination angle is based on previous research and experience of the authors, please see figures. Max, angle is mainly related to the effective geometry of the ball end milling cuter, although with the Solid carbide end mill, CoroMill® Plura, D6 mm we are not limited to the effective tool geometry.
Similar courses (similarity bathtub curve) are recorded (not only our researcher, see figures) in other studies conducted by other authors under different machining conditions (machining, cutting tools, machine tools, cutting conditions, etc). That’s the reason, that we decided to use maximum inclination angle in this samples 20.
New comment was add: [100] Maximum inclination angle was set according to shallow shapes of the sample.
[4] Did you calibrate the machine tool before the experiment? Calibration can make sure the experimental machine tool has a known accuracy state. Then, the original machine tool precision can not bring some affections to the final experiment result.
Yes, we did. We use 3D quickset (DMU 50, 574028) from company Deckel Maho before making experimental machining.
[5] Machine tools have different volumetric errors in the machining space. You need to know the volumetric error information for your machining space. Volumatic errors definitely can affect the comparison results.
Yes, volumetric errors can affect the results, but we try to provide the same condition and same setup for experiments. We try to consider positioning accuracy, interpolation accuracy, volumetric accuracy, and thermal expansion as constant.
Research target is - if the tool inclination has an influence on the accuracy of free form surface machining during constant and stabile conditions (machine tool, holders, fixtures, ball end milling cutter, cutting conditions) We didn't find out accuracy of machine tool.
We use Renishaw's Ballbar QC20-W to determine the condition of the machine tool (Interpolation accuracy) before experimental part.
We also use laser interferometer for verification of positioning accuracy of the machine tool with cooperation of University of Zilina.
[6] It is hard to recognize the detail of Fig. 11- Fig. 20, especially for some statistical parameters. You need to use figures with high solutions.
New figures (Fig. 11 – Fig. 20) with high resolution was add.
[7] Except for the roughness, parameters such as flatness and parallelism could also be selected for this research. You may consider these parameters.
Design of samples were designed with a view to analysing of shape deviations and surface roughness. (We have added only some of the results to the article, see Tab. 3 and Tab.4. Other results will be published in the following article.
No measurement was expected for these test samples of geometrical deviation (Perpendicularity, Parallelism, Flatness and so on...) For these type of research, we have to design to appropriate shape of samples.
Suitable shape of samples for follow-up research will be proposed as your input. Thank you for your suggestion.
[8] Could you please give the reader some explanations on the differences between 3+2 axis matching and 5-axis machining?
We add this explanation [line 96]:
“3+2 axis machining is 3axis machining with fixed toll axis orientation. Simultaneous 5axis machining moves the cutting tool on the x, y and z axes and rotate the A, B and C axes (in our kinematic B and C axes) to maintain continuous contact between the tool and workpiece, unlike 3+2 axis machining, where the part is in a fixed orientation”.
3+2 process is also referred to as positional five-axis machining because the fourth and fifth rotary axes keep the part in a fixed orientation, then typical 3axis machining can be carried out, instead of moving it continuously during the machining process.
[9] Line 219, you mentioned that they are under the same condition, Why? For different setups, the G-code is different, or even the tool path is also different. Does this affect the result?
The same conditions means the same cutting conditions (same fz, vc, ap, ae, …) see Tab. 2. Tool paths are the same. Tool axis inclination angles are different.
New statement was added to the line [249] “…. - same cutting conditions (same fz, vc, ap, ae, …see Tab.2.)“
[10] Line 221-222, What is the result of 3-axis machining? Do you mean that 3-axis machining has a close machining quality as the 3+2 axis machining? This description is not clear. You can use a table to show the roughness of each sample.
We add a new chapter “3.3. Surface rougness measurement”, tables (Tab.3 and Tab.4.) and new descriptions.
Thank you for your valuable comments and for the opportunity to improve our manuscript.
Therefore, we were able to complete the data and prepare better-informed discussion part of the paper. We decided to add the surface roughness data of the central area of samples.

Reviewer 2 Report
The description of state of knowledge is insufficient. There are no conclusions from the literature review. There is no novelty. The authors did not solve any problem.
Author Response
My Response to Reviewer
Dear Reviewer,
Thank you for your letter and the reviewers’ comments concerning our manuscript entitled. These comments are all valuable and very helpful for revising and improving our paper, as well as the important guiding significance to our researches.
We have studied comments carefully and have made correction which we hope meet with the approval. According to the comments, major changes in the revised manuscript are highlighted using the "Track Changes" function in Microsoft Word.
Comments and Suggestions for Authors reviewer 2
The description of state of knowledge is insufficient. There are no conclusions from the literature review. There is no novelty. The authors did not solve any problem.
The missing experiments were realized and added, some evaluation faults were discovered, and the results, discussion and conclusion parts were corrected and completed. We hope that we have improved our paper substantially and that you might find it more interesting and important now.

Reviewer 3 Report
- Lines 47-65 need to address the reference error messages
- In Fig. 7, 5 axis, Series βf=0°, the following 5 boxes, βf=0° should be the same but your final 3 boxes are 5. You can remove all βf in the boxes as it already shown in Series boxes.
- the text boxes within the graphics from Fig 11 to Fig 19 are mostly unreadable -Please use the clear pics.
- I don't understand the following sentences in lines 198-200 'The same result is not in the highest "z" level of the workpiece (on the left of the sample). This prediction of 3+2 milling before the real process is not conclusive in practice. This comparison is therefore rather indicative.'
- Please use a table to summary how many experiments you did. In the Discussion and Conclusions section, the best result is 17um using 5 axis milling with βf = 0 °, βn = 10 ° . Please also compare the result when you use 3+2 axis milling with βf = 0 °, βn = 10 °.
Author Response
My Response to Reviewer
Dear Reviewer,
Thank you for your letter and the reviewers’ comments concerning our manuscript entitled. These comments are all valuable and very helpful for revising and improving our paper, as well as the important guiding significance to our researches.
We have studied comments carefully and have made correction which we hope meet with the approval. According to the comments, major changes in the revised manuscript are highlighted using the "Track Changes" function in Microsoft Word.
Comments and Suggestions for Authors reviewer 3
- Lines 47-65 need to address the reference error messages
Missing reference number has been added. This sentence was corrected and improved: ”Using multiaxis machining (3+2 and 5axis machining) include these benefits, which is based on previous research by the authors [19], [20], [21], [35], [36]”.
- In Fig. 7, 5 axis, Series βf=0°, the following 5 boxes, βf=0° should be the same but your final 3 boxes are 5. You can remove all βf in the boxes as it already shown in Series boxes.
Fig. 7 was improved.
- the text boxes within the graphics from Fig 11 to Fig 19 are mostly unreadable -Please use the clear pics.
New figures (Fig. 11 – Fig. 20) with high resolution was add.
- I don't understand the following sentences in lines 198-200 'The same result is not in the highest "z" level of the workpiece (on the left of the sample). This prediction of 3+2 milling before the real process is not conclusive in practice. This comparison is therefore rather indicative.'
These sentences [line 201] was improved as:“ The prediction of the formation of a planar surface (on the left of the sample) in the CAM system Mastercam does not correspond to the actually measured results”
- Please use a table to summary how many experiments you did. In the Discussion and Conclusions section, the best result is 17um using 5 axis milling with βf = 0 °, βn = 10 ° . Please also compare the result when you use 3+2 axis milling with βf = 0 °, βn = 10 °.
We add a new chapter “3.3. Surface rougness measurement”, tables (Tab.3 and Tab.4.) and new descriptions.
Thank you for all your suggestions and opinions. The missing experiments were realized and added, some evaluation faults were discovered, and the results, discussion and conclusion parts were corrected and completed. We hope that we have improved our paper substantially and that you might find it more interesting and important now.

Reviewer 4 Report
The precision of the surfaces machined on machine tools is a matter of current interest and of great importance.
The work is interesting and the research can be useful in the field of milling processing on numerically controlled machines.
The following issues should be corrected:
- on page 2, lines 47-49; 50-52 etc. references (probably inserted automatically) are not found and the error message "References source not found" appears;
- on page 7, line 131 refers to figure 10 but, I think that in reality it is figure 9.
- on page 11, row 191, in the legend of the figure appears fig. 1 but it is probably figure 18.
Author Response
My Response to Reviewer
Dear Reviewer,
Thank you for your letter and the reviewers’ comments concerning our manuscript entitled. These comments are all valuable and very helpful for revising and improving our paper, as well as the important guiding significance to our researches.
We have studied comments carefully and have made correction which we hope meet with the approval. According to the comments, major changes in the revised manuscript are highlighted using the "Track Changes" function in Microsoft Word.
Comments and Suggestions for Authors reviewer 4
The precision of the surfaces machined on machine tools is a matter of current interest and of great importance.
The work is interesting and the research can be useful in the field of milling processing on numerically controlled machines.
The following issues should be corrected:
- on page 2, lines 47-49; 50-52 etc. references (probably inserted automatically) are not found and the error message "References source not found" appears;
Missing reference number has been added. This sentence was corrected and improved: ”Using multiaxis machining (3+2 and 5axis machining) include these benefits, which is based on previous research by the authors [19], [20], [21], [35], [36]”.
- on page 7, line 131 refers to figure 10 but, I think that in reality it is figure 9.
We correct refers to figure 9.
- on page 11, row 191, in the legend of the figure appears fig. 1 but it is probably figure 18.
We correct to figure 18.
Thank you for all your suggestions and opinions. The missing experiments were realized and added, some evaluation faults were discovered, and the results, discussion and conclusion parts were corrected and completed. We hope that we have improved our paper substantially and that you might find it more interesting and important now.

Round 2
Reviewer 1 Report
The author fully responds to my comments. I am satisfied with their answers. Technically, I think this paper contains the necessary work for the discussed topic. Data are well analyzed and presented.
I have only one concern, the analysis of surface roughness (Line 234-238). Better surface roughness was achieved 3+2axis milling. I do not think this is very clear. The differences between each milling method are 2.05, 1.99 and 2.3, not very big. I may say 3 axis and 3+2axis have close Sa, 5 axis has a weaker performance. In addition, in the conclusion part, there is no clear result related to surface roughness.
Finally, if the introduction part can be improved, that would be perfect.
Author Response
Dear Reviewer,
Thank you for your comment. I really appreciate it, all comments are valuable and very helpful for improving our paper.
Changes in the manuscript are highlighted using the "Track Changes" function in Microsoft Word.
We add [line 237]: “However, the values of the surface roughness are very similar. Within the extended uncertainty of measurement, values are close to each other, in particular the parameter Sa (±0.15 µm and ±0.51 µm).“
We add Extended uncertainty values to the Tab. 3 and Tab. 4.
I removed Fig. 21. because I didn’t receive any answer for copyright permission for this figure from AALICONA Imaging GmbH.
Reviewer 2 Report
I have read the revised version of manuscript and I think this article is not warrants publication in Materials.Author Response
Dear Reviewer,
Thank you for your comment.
We add [line 237]: “However, the values of the surface roughness are very similar. Within the extended uncertainty of measurement, values are close to each other, in particular the parameter Sa (±0.15 µm and ±0.51 µm).“
We add Extended uncertainty values to the Tab. 3 and Tab. 4.
I removed Fig. 21. because I didn’t receive any answer for copyright permission for this figure from AALICONA Imaging GmbH.